# Performance and Safety of a Reflux-Control Microcatheter Used to Perform DEB-TACE with LUMI^TM^ Beads in HCC Patients: Preliminary Experience

**DOI:** 10.3390/jcm12206630

**Published:** 2023-10-19

**Authors:** Salvatore Alessio Angileri, Carolina Lanza, Serena Carriero, Pierpaolo Biondetti, Velio Ascenti, Giuseppe Pellegrino, Alessandro Caruso, Gianpaolo Carrafiello, Anna Maria Ierardi

**Affiliations:** 1Department of Diagnostic and Interventional Radiology, Foundation IRCCS Cà Granda—Ospedale Maggiore Policlinico, Via Francesco Sforza 35, 20122 Milan, Italy; 2Postgraduate School in Radiodiagnostics, Università Degli Studi di Milano, 20122 Milan, Italy; 3School of Medicine, Università Degli Studi di Milano, Via Festa del Perdono 7, 20122 Milan, Italy; alessandrocars97@gmail.com

**Keywords:** anti-reflux microcatheter, DEB-TACE, radiopaque microspheres, hepatocellular carcinoma (HCC), the non-target embolization (NTE)

## Abstract

Purpose: The present study aims to evaluate the effectiveness and safety of the anti-reflux microcatheter during DEB-TACE with DC Bead LUMI^TM^ (radiopaque beads) for the treatment of hepatocellular carcinoma (HCC). Methods: We performed an observational longitudinal prospective monocentric study to analyze all patients with HCC who underwent to DEB-TACE with DC Bead LUMI^TM^ and anti-reflux microcatheter. Technical success, the presence of residual disease, and clinical success were evaluated. The performance of the anti-reflux microcatheter on the basis of the percentage of tumor covered and the non-target embolization (NTE) was also evaluated. Results: Twenty patients underwent DEB-TACE with DC Bead LUMI^TM^ and an anti-reflux microcatheter. Technical success was achieved in all cases. Residual disease in the target tumor was observed in 11/20 (55%) of cases and no residual disease was found in 9/20 (45%) of cases. The clinical response at 1-month follow-up was of PD 4/20 (20%), SD 7/20 (35%), and CR 9/20 (45%). No major complications were recorded, and 10% of cases had minor complications. The distribution of beads on post-procedural CBCT, classified according to the percentage of target nodule coverage, was ≥50% in 70% (14/20) of cases and between 30–50% in 30% of cases (6/20). NTE was never registered.

## 1. Introduction

Trans-arterial chemo-embolization (TACE) belongs to the arterially directed embolic therapies and is a widely used loco-regional therapy for the treatment of advanced or unresectable hepatocellular carcinoma (HCC) [1].

According to the Barcelona Clinical Liver Cancer (BCLC) system, TACE is the first-line treatment for intermediate stage disease (BCLC-B).

The BCLC system also takes into account the use of TACE in very early (BCLC-0) and early (BCLC-A) stage patients in cases where other recommended treatments are not practicable or have failed [2,3] survival for HCC [3].

Recent TACE improvements include the introduction of new types of microspheres [4,5]. These have shown to offer a valid and valuable alternative option in TACE procedures, even if overall drug eluting beads–TACE (DEB-TACE) did not improve overall survival as compared with conventional TACE (cTACE), as reported by Precision Italia and other studies [6,7].

DC Bead LUMI™ (Biocompatibles UK Ltd., Farnham, UK, a BTG group company) are among the new types of embolic agents that have recently been introduced in Europe [4]. They are precisely calibrated and radiopaque microspheres that can be loaded with a chemotherapeutic drug. DC Bead LUMI^TM^ beads contain a covalently bound radiopaque moiety that confers inherent and lasting radiopacity. They are visible under imaging modalities such as computed tomography (CT), fluoroscopy, and cone beam CT (CBCT) [4,8]. Thus, DC Bead LUMI^YM^ combines the advantage of low systemic drug absorption and visualization on post-operative imaging. These proprieties can interfere with the recognition of the residual disease on the post-treatment images; a non-contrast-enhanced CT scan phase has to be carried out in order to differentiate the radiopaque beads and the eventual enhancement of the treated area [7,8].

In the United States, LC Bead LUMI^TM^ are available in two sizes (40–90 μm and 70–150 μm). Beads are packaged in a 10 mL glass vial and contain 2 mL of beads suspended in sterile, phosphate-buffered saline (total volume of approximately 8 mL).

Causes of ineffective treatment with TACE may be related to the incorrect catheterization of the feeding vessels as well as non-target embolization (NTE). NTE may be associated with complications [9], where the focal necrosis of the liver parenchyma is adjacent to the embolized HCC nodule (28%), intralesional (micro)abscess (26%), intralesional hemorrhage (22%), or peritumoral bile duct necrosis (12%) [9,10].

In recent years, anti-reflux catheters have been introduced, and pre-clinical studies have demonstrated their efficacy [11]. SeQure microcatheter (Guebert, Villepinte, France) is characterized by a regular lumen, with side slits at its terminal end with the aim of filtering contrast media, creating a fluid barrier between the vessel and the microcatheter; this system reduces microsphere reflux associated with NTE, while preserving normal antegrade blood flow. The microcatheter side holes are to be fixed in the center of the vessel lumen (centro-luminal position) in order to optimize its effect. These marked differences may affect fluid–particle dynamics during microsphere administration and have a significant impact on tumor-targeting during chemoembolization [12].

The present study aims to evaluate the effectiveness and safety of the anti-reflux microcatheter combined with DC-bead LUMI^TM^ during TACE for the treatment of HCC.

## 2. Materials and Methods

### 2.1. Patients

For this observational monocentric longitudinal prospective study, all patients affected by HCC, diagnosed histologically or on the basis of radiological criteria [13], and consecutively treated with TACE performed with DC Beads LUMI^TM^ (Biocompatibles UK Ltd., Farnham, UK, a BTG group company) and epirubicin (Farmorubicin, Pfizer, Japan Inc., Tokyo, Japan) administered via the anti-reflux microcatheter were included.

All subjects gave their consent for inclusion in the study. The study was conducted in accordance with the Declaration of Helsinki, and the protocol was approved by the Ethics Committee of our Institution (project identification code: 30946).

The reporting of this study conforms to STROBE guidelines.

The indication to treatment was established by a multidisciplinary team during weekly meetings. Patients were hospitalized from at least one day before the procedure to at least one day after. If no complications occurred in the first day after the procedure, patients were discharged.

Inclusion criteria included age > 18 years old; informed consent signed; diagnosed HCC unilobar (not suitable for surgery) or bilobar; performance status (PS) of less than 2; BCLC stage A or B [14,15,16,17], platelets (PTL) not inferior to 50 × 10^9^/L, and life expectancy of longer than 3 months.

The use of an anti-reflux microcatheter during DEB-TACE was necessary in the following situations:-Child–Pugh score from A6 to B8, in this latter case, a selective approach is preferred in order to preserve the functioning hepatic parenchyma.-In cases in which the tumor feeding artery arose from the same branch of the right hepatic artery as the cystic artery or if the tumor was nourished by branches of extra-hepatic arteries (e.g., the gastroduodenal artery, left or right gastric artery, phrenic arteries, intercostal arteries, adrenal arteries, superior mesenteric artery, etc.), preventing the risk of backward spread along the abovementioned vessels.

Exclusion criteria included presence of extra-hepatic disease, severe comorbidities, uncorrectable intolerance to epirubicin and contrast medium, and anatomical factors that hinder the feasibility of the procedure.

Absolute contraindications to TACE include portal vein neoplastic thrombosis or hepato-fugal blood flow and poor performance status (ECOG P2 or greater).

### 2.2. Outcome Measures

The following parameters were collected: the dose of DC Beads LUMI^TM^ injected, the percentage of expected tumor coverage assessed via pre-procedural contrast enhanced (CE) CBCT imaging (classified as P1 75–100%, P2 25–75%, P3 < 25%), the real volume of tumor covered (evaluated with post-procedural unenhanced CBCT imaging and classified as A ≤ 15%, B 15–30%, C 30–50%, D > 50%), previous treatments, fistula with venous system at CE CBCT and/or angiogram, microcatheter size, location of the microcatheter tip at CBCT, vessel diameter in correspondence of the microcatheter tip, and type of contrast medium used.

Technical success was defined as the correct positioning of microcatheter at the planned position, ready for injection therapy.

The presence of residual disease in the target tumor nodule was evaluated as follows: 0—none; 1—presence.

Clinical success was defined as the response to the treatment at 1-month follow-up, and the imaging was evaluated according to modified response criteria in solid tumors (mRECIST) in the whole liver, apart from the target tumor nodule, as follows: complete response (CR), partial response (PR), stable disease (SD), and progressive disease (PD) [18].

Safety was evaluated via the reporting of any adverse effects, according to CIRSE classification [19], which may have occurred during the procedure and up to one month after the procedure, as well as all-cause mortality within 30 days of the procedure.

The performance of the microcatheter was evaluated on the basis of the percentage of tumor covered and was assessed by comparing pre-procedural CE CBCT imaging (revealing the expected volume to cover) with the real volume of tumor covered (evaluated through post-procedural CBCT imaging) and the non-target embolization (NTE).

NTE was evaluated via immediate post-procedural CBCT as follows: 0—no off-target accumulation of beads caused by vessel anomalies (e.g., shunts); 1—present off-target accumulation of beads caused by vessel anomalies (e.g., shunts); 2—present reflux or non-target accumulation of beads caused by backflow (bead reflux).

### 2.3. Procedure Details

In accordance with current clinical practice of our hospital, DEB-TACE was performed by infusing one vial of DC Bead LUMI^TM^ loaded with epirubicin.

A very high dilution that allowed an adequate suspension time, no aggregate formation, and low solution viscosity was used: this careful management aims to avoid the settling of beads in the syringe and the potential clogging of the microcatheter.

Written informed consent was obtained from all patients. All procedures, performed in an angiographic suite using digital subtraction angiography (DSA) and cone–beam CT (CBCT), were performed using the same protocol. Through a 5-Fr transfemoral introducer sheath and a 5-Fr Cobra or Simmons 1 catheter (Cordis, Miami Lakes, FL, USA), the preliminary selective angiography of the celiac trunk and the catheterization of the common hepatic artery was achieved.

In all patients a SeQure^®^ microcatheter (Guebert, France) was used to achieve the super selective micro-catheterization of tumor vessels.

CE CBCT was performed before injecting the beads to verify the correct positioning of the microcatheter and the percentage of the nodule covered from that position.

Unenhanced CBCT was performed immediately after the completion of the procedure in order to confirm bead mapping. To avoid the misinterpretation of the hepatic parenchymal enhancement as a sign of bead accumulation, the CBCT post-TACE was performed at least 5 min after intra-arterial contrast medium injection (Figure 1).

Beads (2 mL DC Bead LUMI^TM^) were mixed with contrast media (Iobitridol 370 mg/mL (Xenetix, Guebert, Villepinte, France) or Iomeprol 400 mg/mL (Iomeron, Bracco, Italy) at a dilution of 1:20, keeping the solution in constant agitation to allow the optimal suspension of the particles. The microcatheter used for the catheterization was 2.4, 2.7, or 2.8 F SeQure (Guerbet, France), and as good practice (suggested by manufacturer’s instructions), it was continuously flushed with contrast media. Administration was selective or super-selective according to the requirements of each specific case. The injection was continuous at a fixed rate of 1 mL/min.

The distribution of LUMI^TM^ beads in the liver reflects the function of the anti-reflux microcatheter and was classified according to the percentage of the target nodule covered based on expectations (P1, P2, P3). The parenchyma in which bead accumulation was evident but not included in the expected area was defined as NTE. NTE was evaluated as described above.

### 2.4. Follow-Up Imaging

Standard imaging follow-up was a quadruple-phase CT (pre-contrast, arterial, portal venous, and delayed venous phases) performed at 1-month post-TACE. Tumor response was evaluated following the RECIST, version 1.1, criteria for HCC [16].

### 2.5. Statistical Analysis

Given the small sample size and the observational nature of the study, only descriptive statistics was obtained for all variables assessed in the study population.

Mean and standard deviation were used for normally distributed variables, mean and interquartile ranges for skewed distributions, and proportions for categorical variables. Whenever relevant, 95% confidence intervals (95%CI) were calculated.

No group comparisons were planned.

## 3. Results

### 3.1. Patient and Lesion Characteristics

Between January and October of 2022, 20 consecutive patients (4 females, 16 males; mean age = 69.14 years; age range = 57–85 years old) with HCC who underwent DEB-TACE with DC Bead LUMI^TM^ and an anti-reflux microcatheter were identified.

The disease characteristics, including etiology of liver disease and Child–Pugh classification, are summarized in Table 1.

Of all patients, 6/20 (30%) of them had not received any previous treatment before TACE with LUMI^TM^ beads, while 14/20 (70%) patients had received previous treatment and are classified according to the type of treatment(s) administered and the status of the lesion, as shown in Table 2.

The mean diameter of the target nodules was 29.07 mm (range 9–80 mm).

The mean number of HCC nodules was 2.18 (range 1–5).

### 3.2. Treatment Data

In 11/20 cases (55%), a 2.7 Fr anti-reflux microcatheter was used; in 2/20 cases (10%), a 2.4 Fr was used; and in 7/20 cases (35%), a 2.8 Fr anti-reflux microcatheter were used.

In 7/20 cases (35%), a partial occlusion of microcatheter tip using contrast medium 370 mg/Iodine/mL (Iobitridol; Xenetix, Guebert, Villepinte, France) was observed. In contrast, no microcatheter occlusion was observed using contrast medium 400 mg/Iodine/mL (Iomeprol; Iomeron, Bracco, Italy) in the remaining 13/20 of cases (65%).

The tip of the microcatheter on CBCT images was in all cases (100%) in the correct hepatic segment in accordance with the location of the target lesion.

The mean vessel diameter in contact with the microcatheter tip was 2.5 mm (range of 1.5 to 3.6 mm).

In all cases, the expected patterns covered 75–100% (P1) of the target nodule.

In accordance with current clinical practice of our hospital, DEB-TACE was performed by infusing one vial (2 mL) of LUMI^TM^ beads loaded with epirubicin.

The mean epirubicin total amount was 40.48 mg (range of 18 to 50 mg).

The diameter of the microspheres was 70–150 microns.

The distribution of beads on post-procedural CBCT was clearly visible due its radiopacity, classified according to the percentage of target nodule coverage, which was ≥ of 50% in 70% (14/20) of cases and between 30–50% in 30% of cases (6/20).

NTE was never registered.

No fistula with venous system at CBCT and/or angiogram were observed.

Technical success was achieved in all cases (100%). 

Nodule location according to the segment and central/peripheral involvement are summarized in Table 3.

### 3.3. Lesion Response Rate and Clinical Success

Contrast-enhanced CT, as the standard imaging follow-up at 1-month post-TACE, was performed for all patients.

Residual disease in the target tumor was observed in 11/20 (55%) of cases, and no residual disease was seen in the remaining 9/20 (45%) of cases. The clinical responses at 1-month follow-up were PD 4/20 (20%), SD 7/20 (35%), and CR 9/20 (45%).

No patients died during the study observation period.

### 3.4. Safety

The safety assessment was based on the quantitative and qualitative recording of the adverse events, classified according to CIRSE classification (Table 4).

The median time of observation of adverse events was 1-month after the procedure.

All patients experienced no major complications, and 2/20 (10%) of patients have minor complications, resulted in one case of fever and nausea immediately after the procedure, which was managed with antiemetic and antipyretic therapies, and one case of abdominal pain, which was treated with pain relief therapy. No cases required prolonged hospitalization.

## 4. Discussion

Our study shows that DEB-TACE performed using SeQure with LUMI^TM^ beads in HCC patients is a safe and effective procedure without major adverse events.

The frequency of major complications after TACE is ~4–6% and includes NTE, liver failure, and hepatic abscesses [9,10,20]. The anti-reflux microcatheter has the advantage of being less likely to cause NTE [21]. This advantage is provided by its shape and morphology [21]. Our study emphasized this advantage by not showing NTE in 100% of cases.

A partial occlusion of the microcatheter tip was observed in 35% of cases using a contrast medium 370 mg/Iodine/mL. This event was not observed using a contrast medium of 400 mg/Iodine/mL. The most likely hypothesis concerning this is that the contrast medium 400 mg/Iodine/mL has a higher viscosity than other LOCMs (low osmolarity contrast agents). This hypothesis needs to be confirmed by further experimentation and future studies.

LUMI^TM^ beads can be visualized in real-time during procedures using fluoroscopic guidance and post-procedural imaging; this provides the interventional radiologist with immediate intraprocedural feedback on the distribution of the beads and local drug delivery [4,5]. SeQure, the microcatheter used for all procedures, has two unique side slits near the main tip that allow fluid to exit parallel to the course of the catheter, resulting in a protective fluid barrier at the sides of the catheter that prevents the unwanted upflow of embolizing material [22].

The combination of radiopaque beads and an anti-reflux microcatheter allows for the early identification of radiopaque beads in non-target areas, preventing or minimizing NTE. In addition, radiopacity also helps radiologists in the interpretation of follow-up CT scans, as it is always possible to observe the area covered by beads [4,5]. In our study, the distribution of LUMI^TM^ beads on post-procedural CBCT, classified according to the percentage of target nodule coverage, was ≥ of 50% in 70% of cases.

DEB-TACE, in comparison with cTACE, as shown, facilitates higher concentrations of drugs within the target tumor and lower systemic concentrations [22]. In some patients, the lipiodol oil emulsion in C-TACE has been associated with severe pain [22]. DEB-TACE reduces the systemic side effects of chemotherapy because it promotes the selective and controlled delivery of cytotoxic drugs [22]. The DEB-TACE permits the release of higher concentrations of drugs in tumor tissues in a controlled and sustained manner, thus reducing systemic concentrations [8,14].

The study’s main limitations were the small sample size, the monocentric design, and the fact that interventional procedures were performed by different operators with different skill levels and years of experience.

The effectiveness of DEB-TACE in HCC is well known; the use of a SeQure microcatheter represents a new possibility of overcoming some of the limits of DEB-TACE.

## 5. Conclusions

In conclusion, in our preliminary experience, the combined use of DC LUMI ^TM^ with a SeQure microcatheter is a safe and effective procedure that may prevent non-target embolization events, potentially increasing the efficacy of the treatment and reducing peri-procedural complications. More numerous series and/or randomized trials would be useful to confirm our data.

## Figures and Tables

**Figure 1 jcm-12-06630-f001:**
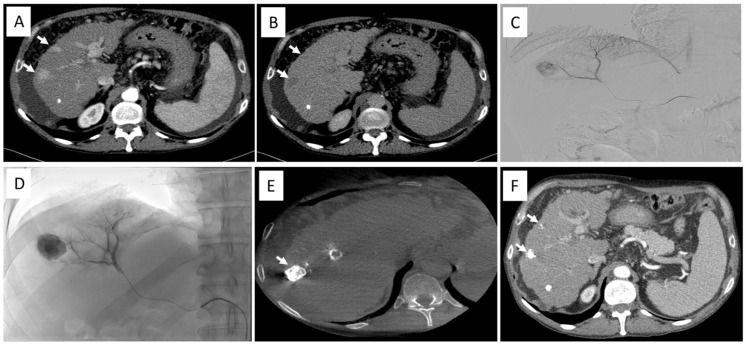
(**A**,**B**) Contrast-enhanced CT scan shows the presence of two nodules of HCC with arterial phase non-rim hyperenhancement and non-peripheral washout (white arrows); (**C**) selective angiography from the right hepatic artery shows the larger nodule with intense hyperenhancement; (**D**) fluoroscopic image during the injection of 400 mg/Iodine/mL contrast medium; (**E**) final CBCT demonstrates the accumulation of radiopaque DC Beads LUMI^TM^ within the larger nodule; (**F**) one-month follow-up CECT shows the presence of radiopaque DC Beads LUMI^TM^ in in accordance with both target nodules with no residual disease. Abbreviations: computed tomography (CT), hepatocellular carcinoma (HCC), cone–beam CT (CBCT), contrast-enhanced CT (CECT).

**Table 1 jcm-12-06630-t001:** The disease characteristics, including etiology of liver disease and Child–Pugh classification. Abbreviations: alcoholic steatohepatitis (ASH); hepatitis B virus (HBV); hepatitis C virus (HCV); non-alcoholic steatohepatitis (NASH); primary biliary cholangitis (PBC).

Patients	Etiology	Child–Pugh Classification
1	ASH/NASH	B7
2	NASH	B7
3	PBC	A6
4	HCV	A6
5	HCV	A5
6	HCV + NASH	A6
7	HCV	B9
8	ASH	A6
9	NASH	B8
10	ASH	A6
11	ASH + HCV	A6
12	ASH	A6
13	ASH	B7
14	HBV + HDV + ASH	A6
15	HCV	A6
16	HCV	B7
17	ASH + HBV	B7
18	HBV	B8
19	ASH/NASH	A5
20	HCV	A6

**Table 2 jcm-12-06630-t002:** Types of previous treatment(s). Abbreviations: trans-arterial chemoembolization (TACE); microwave ablation (MWA); trans-arterial radioembolization (TARE); N/A—not applicable.

Patients	Previous Treatment(s)
1	N/A
2	N/A
3	TACE
4	N/A
5	MWA + TACE
6	MWA
7	MWA
8	TARE
9	MWA + TACE
10	TACE
11	TACE
12	N/A
13	TACE + TARE
14	TACE
15	MWA
16	N/A
17	N/A
18	TACE
19	TACE
20	MWA

**Table 3 jcm-12-06630-t003:** Nodule location according to the segment and central/peripheral involvement.

Patients	Hepatic Lobe	Peripheral/Central
1	S8	Peripheral
2	S2, S4, S5	Peripheral
3	S2	Peripheral
4	S4a-8	Peripheral
5	S6	Peripheral
6	S8	Central
7	S8	Central
8	S5, S6, S8	Peripheral
9	S7, S8	Peripheral
10	S3	Peripheral
11	S8, S4	Peripheral
12	S7	Peripheral
13	S8, S5	Peripheral
14	S8	Central
15	S4b	Peripheral
16	S7	Peripheral
17	S5	Central
18	S4a	Central
19	S8	Peripheral
20	S8-4	Central

**Table 4 jcm-12-06630-t004:** Quantitative and qualitative classification of adverse events, according to CIRSE classification [16].

1	Complication during the procedure which could be solved within the same session; no additional therapy; no postprocedural sequelae; no deviation from the normal post-therapeutic course
2	Prolonged observation including overnight stay (as a deviation from the normal post-therapeutic course <48 h); no additional postprocedural therapy; no postprocedural sequelae
3	Additional postprocedural therapy or prolonged hospital stay (>48 h) required; no postprocedural sequelae
4	Complication causing a permanent mild sequelae (resuming work and independent living)
5	Complication causing a permanent severe sequelae (requiring ongoing assistance in daily life)
6	Death

## Data Availability

The data presented in this study are available on request from the corresponding author.

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
