# Peer review of "Performance and Safety of a Reflux-Control Microcatheter Used to Perform DEB-TACE with LUMITM Beads in HCC Patients: Preliminary Experience"

_jcm, 2023, doi:10.3390/jcm12206630_

Round 1

Reviewer 1 Report

GENERAL COMMENTS

Synopsis: The authors prospectively analyze 20 DEB-TACE treatments performed with radiopaque beads (LUMI-BSC) using an anti-reflux fluid-dynamic control microcatheter (SeQure, Guerbert). The hypothesis of effective anti-reflux effect of the microcatheter tip can be demonstrated because due to the radiopaque nature of the beads it is possible to confirm the deposits of the beads in the target tumor and to exclude the abnormal deposition of the beads in any non-target location.

SeQure® catheter is well known among IR doctors due to the stiffness of this device compared with other type of microcatheteres. Performance assessment of this device should be based on percentage of tumor covered and the presence or not of non-target embolization as authors did. However, SeQure performancee should also include the capability of this device to reach the most distal portion of the hepatic artery without causing spasm or dissection.

If the mean size of target nodules was 29 mm why were these tumors not treated with ablation instead of TACE?  Please clarify

LUMI as well as lipiodol may interfere with the response assessment when it is performed with enhanced  CT. Authors have to clarify how did they overcome this issue.

Please edit in neutral voicing. In scientific language, the correct and polite is to avoid all first person verbiage (we, our…)

Title: OK

ABSTRACT:

Please clarify in the abstract that LUMI beads are radiopaque

Please clarify in the abstract the single center nature of this study.

P1L14: Please remove commercial information from the abstract

KEY WORDS: 

“DC Bead LUMI” is not a Key word. I suggest to change by “radiopaque microspheres”.

INTRODUCTION:

The introduction of this manuscript is large and should be shortened.

1 to 4 paragraphs are well known for the readers of this paper and should be summarize in 2 lines.

P2L45: References 4 and 5 belongs to the same author. To point out the improvements of DEB-TACE it could be useful to cite different authors.

I suggest: J Urbano et al. Multicentre prospective study of drug-eluting bead chemoembolisation safety using tightly calibrated small microspheres in non-resectable hepatocellular. Eur J Radiol . 2020 May;126:108966. doi: 10.1016/j.ejrad.2020.108966.

MATERIAL & METHODS:

Were the patients included in this study consecutive patients?

Please add a brief description of the anti-reflux microcatheter and how it works.

When TACE needs a second session, where the TACE treatments done on demand or scheduled?

Please add a brief description of the LUMI Beads. Size, why are they radiopaque, type and dose of drug load….?

P2L78: add a reference to support the non-invasive criteria of HCC

P3L79: please include manufacturer, brand and country of Lumi. Boston……

P3L80: remove “new”. Sequre is in the market for more than 3 years

P3L85-86: Remove “during ………interventional radiologist”. Redundant information.

P93-107: The indication of anti-reflux microcatheter is quite confusing. Please try to put in a chat or to list in a clear way.  Exclusion criteria of what’ … exclusion criteria of  Sequre application…?

P3L118: This sentence is vague. Do author mean that they checked the complete or incomplete deposition of radiopaque beads along the tumor volume? How was this confirmed? By CBCT?

P3L121: mRECITS needs a reference.

P3L127: Performance of the micro and technical success seems the same. Please clarify if you consider different or same endpoint.

RESULTS:

The size and number of HCC treated have to be clarified. Eg in table 1 or 3

How many cases were done with 2,4, 2.7 and 2.8 F microcatheter. Did authors find any difference in the results regarding the micro size?

Procedure details of the Results need a deep review.

P4L142-156 belong to M&M section.

P5L212: did lumi beads make it difficulty the 1-month control CT scan?

DISCUSSION

No word and no discussion at all about the pros and cons of Sequere catheter.

P9L59: change demonstrates by shows

P9L270-271: repeated sentence, please remove

CONCLUSION: Ok

REFERENCES: OK

TABLES:

Table 1 and 2:  OK

Table 3: No information regarding central or peripheral location

FIGURES AND LEGENDS:

Fig 1: If no wash out (fig 1b) …. Did this patient received a pre-TACE biopsy to confirm the HCC diagnosis?

.

Author Response

GENERAL COMMENTS

Synopsis: The authors prospectively analyze 20 DEB-TACE treatments performed with radiopaque beads (LUMI-BSC) using an anti-reflux fluid-dynamic control microcatheter (SeQure, Guerbert). The hypothesis of effective anti-reflux effect of the microcatheter tip can be demonstrated because due to the radiopaque nature of the beads it is possible to confirm the deposits of the beads in the target tumor and to exclude the abnormal deposition of the beads in any non-target location.

SeQure® catheter is well known among IR doctors due to the stiffness of this device compared with other type of microcatheteres. Performance assessment of this device should be based on percentage of tumor covered and the presence or not of non-target embolization as authors did. However, SeQure performancee should also include the capability of this device to reach the most distal portion of the hepatic artery without causing spasm or dissection.

If the mean size of target nodules was 29 mm why were these tumors not treated with ablation instead of TACE? Please clarify

The size referring to the target nodule(s) mean size, referring to the bigger nodule treated, but some patients have three or more nodules, then, according to the current guidelines and after decision of multidisciplinary board the treatment of choice was TACE.

LUMI as well as lipiodol may interfere with the response assessment when it is performed with enhanced CT. Authors have to clarify how did they overcome this issue.

Thank you for the comment. We have modified the text as follow:

“Those proprieties can interfere with the recognition of the residual disease on the post-treatment images; a non-contrast enhanced CT scan phase has to be done in order to differentiate the radiopaque beads and the eventually enhancement on treated area.” (line57-60)

Please edit in neutral voicing. In scientific language, the correct and polite is to avoid all first person verbiage (we, our…)

Thank you for the comment we have modified as suggested.

Title: OK

ABSTRACT:

Please clarify in the abstract that LUMI beads are radiopaque

Thank you for the comment we have added this information (line 14)

Please clarify in the abstract the single center nature of this study.

Thank you for the comment we have added this information (line 16,80)

P1L14: Please remove commercial information from the abstract

Thank you for the comment we have removed the commercial informations. (line14)

KEY WORDS:

“DC Bead LUMI” is not a Key word. I suggest to change by “radiopaque microspheres”.

Thank you for the comment we have modified as suggested (line 28)

INTRODUCTION: 2

The introduction of this manuscript is large and should be shortened.

1 to 4 paragraphs are well known for the readers of this paper and should be summarize in 2 lines.

Thank you for the comment we have modified as suggested (line3 to 44).

P2L45: References 4 and 5 belongs to the same author. To point out the improvements of DEB-TACE it could be useful to cite different authors.

I suggest: J Urbano et al. Multicentre prospective study of drug-eluting bead chemoembolisation safety using tightly calibrated small microspheres in non-resectable hepatocellular. Eur J Radiol . 2020 May;126:108966. doi: 10.1016/j.ejrad.2020.108966.

Thank you for the comment we have modified as suggested.

MATERIAL & METHODS:

Were the patients included in this study consecutive patients?

Thank you for the comment we have modified as suggested (line 81)

Please add a brief description of the anti-reflux microcatheter and how it works.

Thank you for the comment. The description of the anti-reflux microcatheter was already provided into the introduction section: ….”In the last years, anti-reflux catheters were introduced, and pre-clinical studies have demonstrated their efficacy [11]. SeQure microcatheter (Guebert, Villepinte, France) is characterized by a regular lumen, with side slits at its terminal end with the aim to filter contrast media, creating a fluid barrier between the vessel and the microcatheter; this system reduces microspheres reflux associated with NTE, while preserving normal an-tegrade blood flow. The microcatheter side holes are to be fixed in the center of the vessel lumen (centro-luminal position), in order to optimize its’ effect. These marked differences may affect fluid-particle dynamics during microsphere administrations and have a sig-nificant impact on tumor targeting during chemoembolization.”

When TACE needs a second session, where the TACE treatments done on demand or scheduled?

Thank you for the comment, when the TACE need a second session, it was scheduled.

Please add a brief description of the LUMI Beads. Size, why are they radiopaque, type and dose of drug load….?

Thank you for the comment we have added these informations (lines 50-60)

DC Bead LUMI™ (Biocompatibles UK Ltd, Farnham, UK, a BTG group company) are among the new types of embolic agents that have been recently introduced in Europe [4]. They are precisely calibrated and radiopaque microspheres that can be loaded with a chemotherapeutic drug. DC Bead LUMITM beads contain a covalently bound radiopaque moiety that confers inherent and lasting radiopacity.

In the United States, LC Bead LUMITM are available in two sizes (40-90μm and 70-150μm). Beads are packaged in a 10mL glass vial, and contain 2mL of beads suspended in sterile, phosphate-buffered saline (total volume of approximately 8mL). [7,8]

We have used as chemiotherapic agent the epirubicin and the details of dose of drug in into the results section (line 208).

“…The mean epirubicin total amount was 40,48 mg (range 18 and 50 mg).”

P2L78: add a reference to support the non-invasive criteria of HCC

Thank you for the comment we have added the reference n. 13

P3L79: please include manufacturer, brand and country of Lumi. Boston……

Thank you for the comment we have added the informations

P3L80: remove “new”. Sequre is in the market for more than 3 years

Thank you for the comment we have modified as suggested

P3L85-86: Remove “during ………interventional radiologist”. Redundant information.

Thank you for the comment we have modified as suggested

P93-107: The indication of anti-reflux microcatheter is quite confusing. Please try to put in a chat or to list in a clear way. Exclusion criteria of what’ … exclusion criteria of Sequre application…?

Thank you for the comment, the indications of the anti-reflux microcatheter usage were listed as follow:

The use of anti-reflux microcatheter during DEB-TACE was indicated in the following situations:

- Child-Pugh Score from A6 to B8, in this latter case a selective approach is preferred in order to preserve the functioning hepatic parenchyma.

- In cases in which the tumor feeding artery arises from the same branch of the right hepatic artery as the cystic artery or if the tumor is nourished by branches of extra-hepatic arteries (e.g. the gastroduodenal artery, left or right gastric artery, phrenic arteries, intercostal arteries, adrenal arteries, superior mesenteric artery etc.), preventing the risk of spread backward along the abovementioned vessels.

The exclusion criteria for the usage of anti-reflux microcatheter are represented by the exclusion criteria for TACE and are presence of extra-hepatic disease, severe comorbidities, uncorrectable intolerance to epirubicin and contrast medium, anatomical factors that hinder the feasibility of the procedure.  Absolute contraindications to TACE include portal vein neoplastic thrombosis or hepato-fugal blood flow, poor performance status (ECOG P2 or greater). 

P3L118: This sentence is vague. Do author mean that they checked the complete or incomplete deposition of radiopaque beads along the tumor volume? How was this confirmed? By CBCT?

Thank you for the comment: we have modified as follow: “… the percentage of expected tumor coverage assessed by pre-procedural contrast en-hanced (CE) CBCT imaging (classified as P1 75-100%, P2 25-75%, P3<25%), the real volume of tumor covered covered (evaluated with post-procedural unhenhanced CBCT imaging and classified as A</=15%, B 15-30%, C 30-50%, D >50%)” (line 113-117)

P3L121: mRECITS needs a reference.

Thank you for the comment, the reference was the 18.

P3L127: Performance of the micro and technical success seems the same. Please clarify if you consider different or same endpoint.

Thank you for the comment. We have modified as follow:

“Technical success was defined as the correct positioning of microcatheter at the planned position, ready to inject therapy.

The presence of residual disease in the target tumor nodule was evaluated as follows: 0-none, 1-presence.”

RESULTS:

The size and number of HCC treated have to be clarified. Eg in table 1 or 3 3

Thank you for the comment we have added these informations:

The mean diameter of the target nodules was of 29.07 mm (range 9 - 80 mm).

The mean number of HCC nodules was 2,18 (range 1-5)

How many cases were done with 2,4, 2.7 and 2.8 F microcatheter. Did authors find any difference in the results regarding the micro size?

Thank you for the comment. These information were in line 195-196 “ In 11/20 cases (55%) a 2.7 Fr antireflux microcatheter was used, in 2/20 cases (10%) a 2.4 Fr and in 7/20 cases (35%) a 2.8 Fr antireflux microcatheter were used.”

We have not found any differences between the microcatheters used.

Procedure details of the Results need a deep review.

Thank you for the comment, we have modified the text as follow:

“Written informed consent was obtained from all patients. All procedures, performed in an angiographic suite using a digital subtraction angiography (DSA) and a cone-beam-CT (CBCT), were performed with the same protocol.  Through a 5-Fr trans-femoral introducer sheath and a 5-Fr Cobra or Simmons 1 catheter (Cordis, Miami Lakes, Florida), a preliminary selective angiography of the celiac trunk and a catheterization of the common hepatic artery was achieved.

In all patients a SeQure® microcatheter (Guebert, France) was used to achieve a super-selective micro-catheterization of tumor vessels.

CE CBCT was performed before injecting the beads to verify the correct positioning of the microcatheter and the percentage of the nodule covered from that position.

An unhenhanced CBCT was performed immediately after the completion of the procedure in order to check beads mapping. To avoid the misinterpretation of the hepatic parenchymal enhancement as a sign of beads accumulation, the CBCT post-TACE was performed at least 5 minutes after intra-arterial contrast medium injection. (Figure 1)

P4L142-156 belong to M&M section.

Thank you for the comment. 142-156 is already in M&M section.

P5L212: did lumi beads make it difficulty the 1-month control CT scan?

Thank you for the comment. We have modified the text as follow:

“Those proprieties can interfere with the recognition of the residual disease on the post-treatment images; a non-contrast enhanced CT scan phase has to be done in order to differentiate the radiopaque beads and the eventually enhancement on treated area.” (line57-60)

DISCUSSION

No word and no discussion at all about the pros and cons of Sequere catheter.

Thank you for the comment we have modified as follow:

“ SeQure, the microcatether used for all procedure, has two unique side slits near the main tip that allow fluid to exit parallel to the course of the catheter, resulting in a protective fluid barrier at the sides of the catheter that prevents unwanted upflow of embolysing material.[22]

The combination of radiopaque beads and anti-reflux microcatheter allows the  early identification of radiopaque beads in non-target areas preventing or minimizing NTE.”

P9L59: change demonstrates by shows

Thank you for the comment. We have modified as suggested.

P9L270-271: repeated sentence, please remove

Thank you for the comment. We have modified as suggested.

CONCLUSION: Ok

REFERENCES: OK

TABLES:

Table 1 and 2: OK

Table 3: No information regarding central or peripheral location

FIGURES AND LEGENDS:

Fig 1: If no wash out (fig 1b) …. Did this patient received a pre-TACE biopsy to confirm the HCC diagnosis?

Thank you for the comment. The nodule has non-peripheral wash-out, then the diagnosis was made only by radiological assessment.

Reviewer 2 Report

Performance and safety of a reflux-control microcatheter used to perform DEB-TACE with LUMITM beads in HCC patients

After delivering the necessary general information on the use of TACE in HCC it describes the importance of anti-reflux catheters.

It is a observational longitudinal prospective study on a limited number of patients but as preliminary results with some priorities it still is a paper of interest.

The whole research was conducted very thoroughly, with promising results. The technical details may inspire further researchers.

The presentation is clear, comprehensive and well documented.

The references are appropriate, up-to-date and contain 22 titles.

The self-citations only reflect the interest of the authors for related topics.

The figures (6) are supportive for the scope of the paper. The 4 tables present the essential information.

The discussions and conclusions are coherent and connected to the content.

In my opinion the paper fits the journal and the language is correct and understandable.

Author Response

Performance and safety of a reflux-control microcatheter used to perform DEB-TACE with LUMITM beads in HCC patients After delivering the necessary general information on the use of TACE in HCC it describes the importance of anti-reflux catheters. It is a observational longitudinal prospective study on a limited number of patients but as preliminary results with some priorities it still is a paper of interest.

4 The whole research was conducted very thoroughly, with promising results.

The technical details may inspire further researchers.

The presentation is clear, comprehensive and well documented. The references are appropriate, up-to-date and contain 22 titles.

The self-citations only reflect the interest of the authors for related topics.

The figures (6) are supportive for the scope of the paper.

The 4 tables present the essential information. The discussions and conclusions are coherent and connected to the content. In my opinion the paper fits the journal and the language is correct and understandable.

Thank you very much for the encouraging comments.